# Did the COVID-19 pandemic affect levels of burnout, anxiety and depression among doctors and nurses in Bangladesh? A cross-sectional survey study

Hayley Anne Hutchings [1], Mesbah Rahman,[2] Kymberley Carter [1], Saiful Islam,[1] Claire O'Neill,[1] Stephen Roberts,[1] Ann John [1], Greg Fegan,[1] Umakant Dave,[2] Neil Hawkes,[3] Faruque Ahmed,[4] Mahmud Hasan,[5] Abul Kalam Azad,[6] Md Mujibur Rahman,[6] Md Golam Kibria,[7] M Masudur Rahman,[8] Titu Mia,[9] Mahfuza Akhter,[9] John G Williams[10]

For numbered affiliations see end of article.

**Correspondence to**
Professor Hayley Anne Hutchings;
h.a.hutchings@swansea.ac.uk

## ABSTRACT

**Introduction** COVID-19 has caused severe disruption to clinical services in Bangladesh but the extent of this, and the impact on healthcare professionals is unclear. We aimed to assess the perceived levels of anxiety, depression and burnout among doctors and nurses during COVID-19 pandemic.

**Methods** We undertook an online survey using RedCap, directed at doctors and nurses across four institutions in Bangladesh (The Sheikh Russel Gastro Liver Institute & Hospital (SRNGIH), Dhaka Medical College Hospital (DMCH), Mugda Medical College Hospital (MMCH) and M Abdur Rahim Medical College (MARMC) Hospital). We collected information on demographics, awareness of well-being services, COVID-19-related workload, as well as anxiety, depression and burnout using two validated questionnaires: the Hospital Anxiety and Depression Scale (HADS) and the Maslach Burnout Inventory (MBI).

**Results** Of the 3000 participants approached, we received responses from 2705 (90.2%). There was a statistically significant difference in anxiety, depression and burnout scores across institutions (p<0.01). Anxiety, depression and burnout scores were statistically worse in COVID-19 active staff compared with those not working on COVID-19 activities (p<0.01 for HADS anxiety and depression and MBI emotional exhaustion (EE), depersonalisation (DP) and personal accomplishment (PA)). Over half of the participants exhibited some level of anxiety (SRNGIH: 52.2%; DMCH: 53.9%; MMCH: 61.3%; MARMC: 68%) with a high proportion experiencing depression (SRNGIH: 39.5%; DMCH: 38.7%; MMCH: 53.7%; MARMC: 41.1%). Although mean burnout scores were within the normal range for each institution, a high proportion of staff (almost 20% in some instances) were shown to be classified as experiencing burnout by their EE, DP and PA scores.

**Conclusion** We identified a high prevalence of perceived anxiety, depression and burnout among doctors and nurses during the COVID-19 pandemic. This was worse in staff engaged in COVID-19-related activities. These findings

## STRENGTHS AND LIMITATIONS OF THE STUDY

⇒ We conducted a large-scale cross-sectional survey using RedCap, across four institutions in Bangladesh with nurses and doctors.
⇒ We employed two validated measures to assess anxiety, depression and burnout.
⇒ The study was operationalised by personnel based in Bangladesh, while the analysis was conducted remotely in the UK.
⇒ Further work is needed to determine whether anxiety, depression and burnout persist post COVID-19.
⇒ We did not explore whether differences in staff characteristics and awareness of well-being services across institutions had any effect on levels of anxiety, depression and burnout.

could help healthcare organisations to plan for future similar events.

## INTRODUCTION

Bangladesh is classified as a low and/or middle-income country. It has a population of 171 million people and a population density of 1115.62 people per square kilometre, which ranks 10th in the world.[1] The health system of Bangladesh is a pluralistic system with four key sectors: government, private sector, non-governmental organisations and donor agencies.[2] Bangladesh's economy has been growing with a Gross Domestic Product (GDP) growth rate of more than 7.5%; however, a high proportion of the population (about 20%) are living below the poverty line.[3 4] Bangladesh has only 5.3 doctors per 10 000 people, 0.3 nurses per 1000 people, 0.87 hospital beds per 1000 people, 0.72

intensive care unit beds and 1.1 ventilators per 100 000 people.[3]

On 30 January 2020, the WHO declared the outbreak of COVID-19 a public health emergency of international concern.[5] According to the WHO COVID-19 was likely to have the biggest impact in countries with vulnerable healthcare systems. Due to its high population density, poor healthcare systems, poverty and a weak economy Bangladesh had the potential to be at risk of poor outcomes from COVID-19.[3] Bangladesh reported their first case of COVID-19 on 8 March 2020.[3 6 7]

The concept of 'staff burnout' was identified in 1974 by Freudenberger.[8] It was defined as 'burnout in the individual's energy due to failure, wear-out, overloading and unfulfilled desires'. In 2019, 'burnout' was recognised by the WHO as an 'occupational phenomenon',[9] with a dedicated entry in its International Classification of Diseases Handbook (ICD-11 for Mortality and Morbidity Statistics (who.int)).

Since the beginning of the COVID-19 pandemic there has been extensive research exploring the impact of the pandemic on stress, anxiety, depression and burnout within the healthcare community.[10] Researchers in Turkey, Saudi Arabia, USA, Canada, Romania, UK and Portugal have demonstrated the negative impact that COVID-19 has had on levels of burnout, stress and anxiety.[11–18] Factors such as worries about legal issues, age, experience, longer working hours, risk of exposing family members to COVID-19, direct COVID-19 facing roles, being redeployed, gender and coping strategies were related to levels of burnout and stress.

COVID-19 has caused severe disruption in Bangladesh. This has included the country's finances, the delivery of clinical services and its impact on the people of Bangladesh.[19–23] Research in other countries has documented the impact of COVID-19 on healthcare workers, but most of this has been undertaken in higher-income countries.[11–18]

The potential impact that burnout could have from the Bangladesh perspective has previously been considered. Factors including time, pressure, workload and family issues have been identified as being important.[24] Research exploring work stresses in nurses identified that depression was positively associated with workplace violence, bullying and burnout.[25] Female gender, increased social media use, low levels of optimism and sleeping problems increased the risk of depression in trainee medical students during the pandemic in Bangladesh.[26] There has however been limited large-scale work exploring the impact of COVID-19 on anxiety, depression and burnout in healthcare workers in Bangladesh. The aim of this study was therefore to explore levels of burnout, anxiety and depression in doctors and nurses within Bangladesh during the COVID-19 pandemic. We hypothesised that those staff involved in COVID-19 frontline activities would have worse burnout, anxiety and depression than those staff not engaged in COVID-19 frontline activities.

## MATERIALS AND METHODS

### Study sites

A cross-sectional survey study was initiated between June and August 2021 in four large institutions within Bangladesh. We had three sites in Dhaka and one in Dinajpur (northern Bangladesh): The Sheikh Russel Gastro Liver Institute & Hospital (SRNGIH), Dhaka Medical College Hospital (DMCH), Mugda Medical College Hospital (MMCH) and M Abdur Rahim Medical College (MARMC) Hospital.

### Study survey

The survey sought information on personal well-being and current awareness of local well-being services from doctors and nurses working across all specialties at the four institutions. The total number of healthcare staff across the four institutions was 3000 at the time of the survey. The study team included a local clinician at each of the four sites who acted as the site principal investigator (PI). Each PI sent an email on behalf of the study team to all clinical staff at their institution inviting them to participate.

We used validated questionnaires to measure stress and anxiety. We used the Hospital Anxiety and Depression Scale (HADS)[27] to measure anxiety and depression. The HADS has been previously demonstrated to have good reliability, with Cronbach's alpha values between 0.68 and 0.93 for the anxiety and 0.67 and 0.9 for the depression subscales, respectively.[28] A higher score on the HADS indicates a worse anxiety/depression level. In terms of score interpretation, a HADS score of 0–7 is regarded as normal, 8–10 mild, 11–14 moderate and 15–21 severe.[27]

We used the Maslach Burnout Inventory (MBI)[29] for Healthcare staff[30] to measure burnout in the form of emotional exhaustion (EE), depersonalisation/loss of empathy (DP) and personal accomplishment (PA). Reliability of the MBI in healthcare staff has been demonstrated to be acceptable (>0.7 for all subscales).[31 32] Higher scores on the EE and DP subscales of the MBI indicate a higher burnout symptom burden; lower scores on the PA subscale indicate a higher burnout symptom burden.[33] We used previously published MBI raw score cut-off points to assess burnout: ≥31, ≥14 and ≤29, respectively for the EE, DP and PA subscales.[33]

In addition to the two validated questionnaires, we asked participants at which institution they were based, their job/profession, their gender, their age, whether they were working on a COVID-19 rota, whether they were a government employee, whether they undertook any private work and whether they were aware of any well-being services available to them. We also asked them if they had any difficulties answering any of the questions on the HADS or MBI. We provided participant information sheets and consent forms in Bengali and the HADS and MBI in English. Although there was a validated Bengali version of the HADS, there was no Bengali translation for the MBI. Therefore as most of the participants were English speaking, we opted to use English versions

of both questionnaires. We provided additional guidance in Bengali to help participants complete these surveys (if required) and the local PI was available at each institution to provide further guidance, if necessary.

We invited doctors and nurses working both in frontline COVID-19 roles (ie, those directly managing COVID-19 patients) and non-frontline COVID-19 roles to participate in the survey in order for comparisons to be made. Once participants had consented to take part, we provided the survey either on paper or as a link to an online survey via email. We collated the responses and they were entered onto an electronic data capture system REDCap by the study coordinator in Bangladesh. The REDCap database was hosted in Swansea and data analysis was performed by a Swansea-based Bangladeshi statistician.

### Public and patient involvement
None

### Analysis
For the validated surveys, we calculated scores and used the standard processes to manage missing data in line with developer's recommendations.[27 34 35] We explored the completeness of the data from all the surveys and sites and investigated any systematic errors and biases. We looked for evidence of data being missing completely at random. In the analysis, the number and percentages of missing data are provided for all the outcomes.

The HADS anxiety and depression scores and the subscales of the MBI (EE, DP and PA) are presented using summary statistics (mean and SD).

We used one-way analysis of variance (ANOVA) F-tests to compare anxiety, depression and burnout scores across the different institutions. We used Tukey post hoc tests to identify where significant differences existed between different groups after the ANOVA test. We used unpaired t-tests and $\chi^2$ tests to compare scores from participants who actively worked on COVID-19 frontline activities versus non-COVID-19 frontline activities. A p value of <0.05 was regarded as significant. $\chi^2$ tests were used to determine if there was any difference in the proportion of participants reaching threshold levels for anxiety, burnout and depression between institutions. Post-hoc tests with Bonferroni corrections were employed to identify where (if any) the differences existed between institutions. Analysis of the data was carried out using Stata.

### RESULTS
Two thousand seven hundred and five participants out of 3000 (90.2%) responded to our survey. Of the 2705 responses, we received 400 (14.8%) from SRNGIH, 1283 (47.4%) from DMCH, 628 (23.2%) from MMCH and 394 (14.6%) from MARMC. In total, 2108/2705 (77.9%) of the responses were from females and 592/2705 (21.9%) from males, with 5 (0.2%) responses where gender was not specified. Table 1 illustrates the demographic characteristics across the four institutions.

Table 2 illustrates the HADS scores across the four institutions. There was a statistically significant difference in HADS scores across the different institutions (p<0.05). The mean anxiety subscale score was in the normal range (0–7) for two institutions (DMCH and MMCH) and in the mild range (8–10) in the other two institutions. In terms of the mean depression subscale score three of the four institutions had depression scores within the normal range (SRNGIH, DMCH and MARMC), with the fourth institution (MMCH) having a mean score just above the normal range. Figure 1 illustrates the proportion of participants at each institution with normal (0–7), mild (8–10) or abnormal (11–14 moderate; 15–21 severe) HADS anxiety and depression scores. Over half of the participants exhibited some level of anxiety (SRNGIH: 52.2%; DMCH: 53.9%; MMCH: 61.3%; MARMC: 68%) with a high proportion experiencing depression (SRNGIH: 39.5%; DMCH: 38.7%; MMCH: 53.7%; MARMC: 41.1%). There was a statistically significant difference in the proportion of participants experiencing anxiety or depression across the different institutions (p<0.05).

Table 2 illustrates the Maslach scores across the four institutions. There was a statistically significant difference in scores across the four institutions (p<0.05). Mean EE, DP and PA scores did not indicate burnout. Figure 2 illustrates the proportion of staff at each institution who had scores that were classified as reaching the cut-off points that are classified as defining burnout for EE, DP and PA scores.[33] Therefore, although mean scores were within the normal range for each institution, a high proportion of staff (almost 20% in some instances) were shown to be classified as experiencing burnout by their Maslach EE, DP and PA scores. There was a statistically significant difference (p<0.05) in the proportion of participants by institution experiencing burnout as indicated by their EE and PA scores. There was however no significant difference in scores across institutions by DP scores (p>0.05).

As we expected the institution with the highest proportion of COVID-19 rota working (MMCH: 97%) had the highest HADS anxiety and depression scores. MMCH also had the highest MBI scores for the EE and DP subscales of all the institutions and the lowest MBI scores for PA, indicating higher levels of burnout as well as the most individuals achieving these burnout scores.

Table 3 illustrates the impact of a COVID-19 frontline role on HADS and Maslach scores. Healthcare professionals who were engaged with frontline COVID-19 duties had a statistically significantly higher anxiety (8.3 vs 7.8; p<0.01) and depression (7.1 vs 6.5; p<0.05) scores when compared with healthcare professionals who were not engaged with COVID-19 frontline duties. Similarly, Maslach scores for mean EE scores (17.5 vs 15.3; p<0.05) and DP scores (4.9 vs 4.1; p<0.01) were statistically significantly higher in staff with frontline COVID-19 duties compared with staff not in frontline COVID-19 duties, indicating more burnout. Scores for Maslach PA were lower (37.3 vs 39.2; p<0.01) in COVID-19 duty staff than non-COVID-19 duty staff, again indicating worse burnout.

**Table 1** Demographic information by institution

| | Institution | | | |
|---|---|---|---|---|
| | **SRNGIH**<br>**n=400** | **DMCH**<br>**n=1283** | **MMCH**<br>**n=628** | **MARMC**<br>**n=394** |
| Age (years) | | | | |
| Mean (SD) | 32.2 (7.3) | 33.6 (8.4) | 37.1 (8.1) | 32.3 (7.4) |
| Gender (%) | | | | |
| Female:male | 72:28 | 82:18 | 71:29 | 83:17 |
| Departments (n, %) | | | | |
| Internal medicine | 12 (3) | 227 (17.7) | 99 (15.7) | 57 (14.5) |
| Cardiology | 1 (0.3) | 1 (0.1) | 12 (1.9) | 26 (6.6) |
| Gastroenterology | 207 (51.8) | 6 (0.5) | 9 (1.4) | 0 |
| Respiratory medicine | 4 (1.0) | 7 (0.6) | 8 (1.3) | 0 |
| Nephrology | 1 (0.3) | 55 (4.3) | 20 (3.2) | 16 (4.1) |
| Neurology | 0 | 25 (1.9) | 2 (0.3) | 9 (2.3) |
| Rheumatology | 3 (0.8) | 3 (0.2) | 5 (0.8) | 1 (0.3) |
| Surgery | 21 (5.3) | 280 (21.8) | 69 (11.0) | 83 (21.1) |
| Gynae/obs | 1 (0.3) | 124 (9.7) | 57 (9.1) | 38 (9.6) |
| Psychiatry | 1 (0.3) | 8 (0.6) | 4 (0.6) | 0 |
| Other | 149 (37.3) | 533 (41.5) | 340 (54.1) | 156 (39.6) |
| Nmiss (%) | 0 | 14 (1.1) | 3 (0.5) | 8 (2.0) |
| Professions (n, %) | | | | |
| Academic | 31 (7.8) | 2 (0.2) | 119 (18.9) | 5 (1.3) |
| Doctor | 89 (22.3) | 157 (12.2) | 193 (30.7) | 76 (19.3) |
| Nurse | 277 (69.3) | 1092 (85.1) | 311 (49.5) | 312 (79.2) |
| Trainee | 3 (0.8) | 30 (2.3) | 4 (0.6) | 1 (0.3) |
| Nmiss (%) | 0 | 2 (0.2) | 1 (0.2) | 0 |
| COVID-19 rota (n, %) | | | | |
| Yes | 357 (89.3) | 490 (38.2) | 609 (97.0) | 201 (51.0) |
| No | 43 (10.8) | 792 (61.7) | 19 (3.0) | 193 (49.0) |
| Government employee | | | | |
| Yes | 395 (98.8) | 1146 (89.3) | 624 (99.4) | 386 (98.0) |
| No | 5 (1.3) | 137 (10.7) | 3 (0.5) | 8 (2.0) |
| Nmiss (%) | 0 | 0 | 1 (0.2) | 0 |
| Private practice | | | | |
| Yes | 76 (19.0) | 29 (2.3) | 117 (18.6) | 8 (2.0) |
| No | 324 (81.0) | 1252 (97.6) | 510 (81.2) | 86 (98.0) |
| Awareness of a well-being service | | | | |
| Yes | 190 (47.5) | 606 (47.2) | 244 (38.9) | 174 (44.2) |
| No | 209 (52.3) | 672 (52.4) | 383 (60.9) | 214 (54.3) |
| Well-being service (Getting help) (multiple response) | | | | |
| Friends/family | 20 (10.5) | 63 (10.4) | 48 (19.7) | 4 (2.3) |
| Work colleagues | 93 (49.0) | 356 (58.8) | 110 (45.1) | 120 (69.0) |
| Occupational health | 10 (5.3) | 15 (2.5) | 5 (2.1) | 14 (8.1) |
| None | 66 (34.7) | 172 (28.4) | 84 (34.4) | 38 (21.8) |
| Other | 2 (1.1) | 9 (1.5) | 0 | 0 |

Nmiss = number of missing data items
DMCH, Dhaka Medical College Hospital; MARMC, M Abdur Rahim Medical College; MMCH, Mugda Medical College Hospital; SRNGIH, The Sheikh Russel Gastro Liver Institute & Hospital.

**Table 2** Hospital Anxiety and Depression Scale (HADS) scores and Maslach Burnout Inventory scores by institution

| | Institutions | | | | | |
|---|---|---|---|---|---|---|
| **HADS** | **SRNGIH** | **DMCH** | **MMCH** | **MARMC** | **F-test (p value)** | Tukey t-test (significant) |
| Anxiety Subscale Score | | | | | | |
| Mean (SD) | 7.6 (4) | 7.8 (4) | 8.6 (4.1) | 9 (4.3) | 14.7 (<0.01*) | G1 vs G2<br>G1 vs G3*<br>G1 vs G4*<br>G2 vs G3*<br>G2 vs G4*<br>G3 vs G4 |
| Depression Subscale Score | | | | | | |
| Mean (SD) | 6.3 (4.1) | 6.6 (3.7) | 7.9 (3.9) | 6.9 (4.5) | 17.3 (<0.01*) | G1 vs G2<br>G1 vs G3*<br>G1 vs G4<br>G2 vs G3*<br>G2 vs G4<br>G3 vs G4* |
| **Maslach Subscale Scores** | **SRNGIH** | **DMCH** | **MMCH** | **MARMC** | **F-test (p value)** | **Tukey t-test (significant)[1]** |
| Emotional exhaustion (EE) | | | | | | |
| Mean (SD) | 17.5 (11.1) | 15.4 (10.0) | 19.1 (10.8) | 15.7 (8.1) | 20.6 (p<0.01*) | G1 vs G2*<br>G1 vs G3<br>G1 vs G4<br>G2 vs G3*<br>G2 vs G4<br>G3 vs G4* |
| Depersonalisation/loss of empathy (DP) | | | | | | |
| Mean (SD) | 4.3 (5.3) | 4.3 (5.1) | 5.2 (5.2) | 4.7 (5.8) | 4.11 (p<0.01*) | G1 vs G2<br>G1 vs G3*<br>G1 vs G4<br>G2 vs G3*<br>G2 vs G4<br>G3 vs G4 |
| Personal accomplishment (PA) assessment | | | | | | |
| Mean (SD) | 39.0 (8.4) | 38.5 (8.8) | 36.5 (9.4) | 37.8 (7.4) | 8.9 (p<0.01*) | G1 vs G2<br>G1 vs G3*<br>G1 vs G4<br>G2 vs G3*<br>G2 vs G4<br>G3 vs G4 |

* significant difference with p <0.05
DMCH, Dhaka Medical College Hospital; MARMC, M Abdur Rahim Medical College; MMCH, Mugda Medical College Hospital; SRNGIH, The Sheikh Russel Gastro Liver Institute & Hospital.

## DISCUSSION

In this cross-sectional survey, we found that a high proportion of staff working in the four hospitals had some level of anxiety (greater than 50% of participants) and depression (almost 40% of participants). Although the mean MBI scores for the EE, DP and PA scales were within the normal range, a high proportion of participants exceeded the thresholds defined as indicating burnout.[33] There were statistically significant differences across the four institutions for the HADS and MBI scores. We also found that those participants working on COVID-19 activities had worse anxiety, depression and burnout scores than participants who were not working on COVID-19-related activities. The differences in scores across institutions could be explained by the numbers of participants in each institution who were working on COVID-19 activities, as MMCH had the worse anxiety, depression and burnout scores and had the highest number of COVID-19 active staff.

The prevalence of stress, anxiety and depression has been measured in previous studies undertaken throughout the pandemic period. Our findings concur

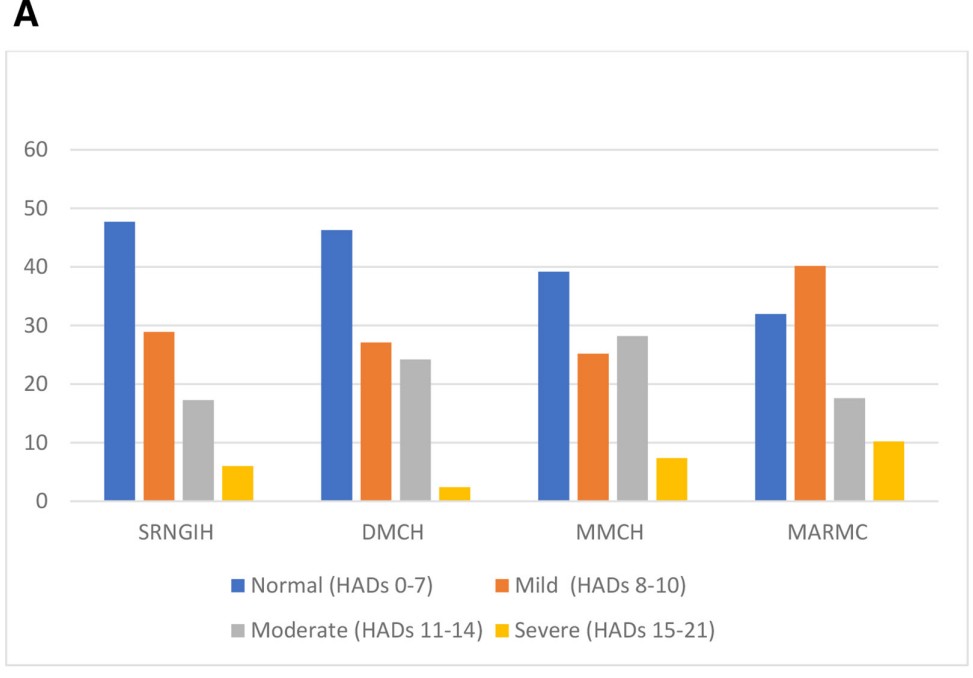

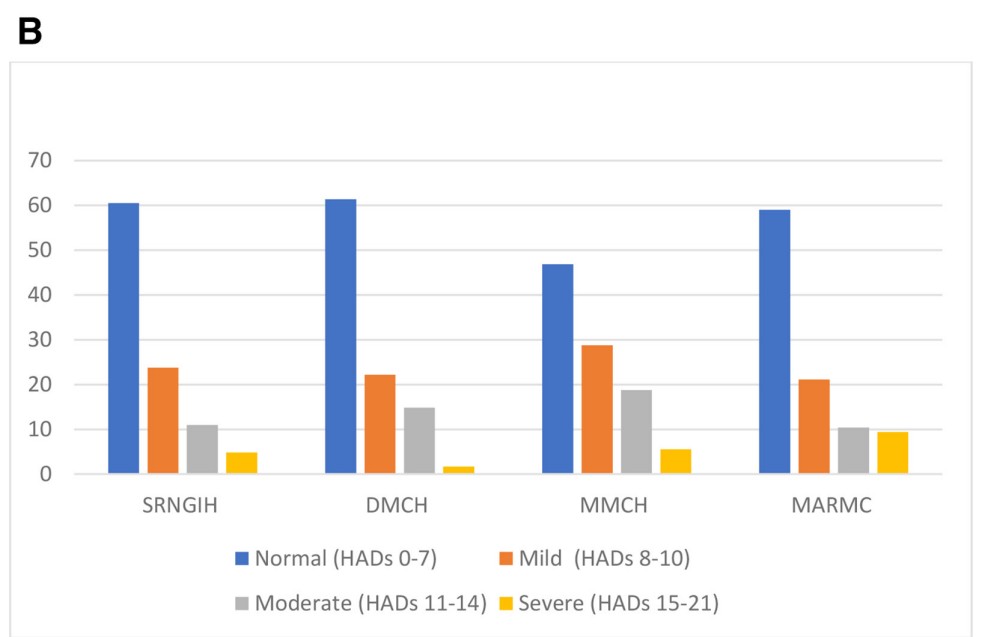

**Figure 1** Proportion of participants exhibiting normal or abnormal HADS anxiety and depression scores by institution. (A) HADS anxiety scores. Statistically significant results ($\chi^2$ comparison with Bonferroni corrections): SRNGIH vs DMCH, p<0.001; SRNGIH vs MMCH, p<0.001; SRNGIH vs MARMC, p<0.001; DMCH vs MMCH, p<0.001; DMCH vs MARMC, p<0.001; MMCH vs MARMC, p<0.001. (B) HADS depression scores. *Statistically significant results ($\chi^2$ comparison with Bonferroni corrections): SRNGIH vs DMCH, p=0.002; SRNGIH vs MMCH, p<0.001; DMCH vs MMCH, p<0.001; DMCH vs MARMC, p<0.001; MMCH vs MARMC, p<0.001. DMCH, Dhaka Medical College Hospital; HADS, Hospital Anxiety and Depression Scale; MARMC, M Abdur Rahim Medical College; MMCH, Mugda Medical College Hospital; SRNGIH, The Sheikh Russel Gastro Liver Institute & Hospital.

with these studies where all have consistently identified a high prevalence of perceived stress, anxiety and depression among healthcare workers.[36 37] A systematic umbrella review reported that there were differences in the prevalence of anxiety and depression among different healthcare workers, but that the prevalence was at least 17% and up to 40% in some groups.[38] As with our findings,

frontline workers appear to be most affected by anxiety and depression.[12]

A study in Bangladeshi nurses during the pandemic showed that depression was linked with burnout and that longer working hours contributed to higher depression.[25] The increased workload associated with the pandemic has also been shown to result in higher levels of depression

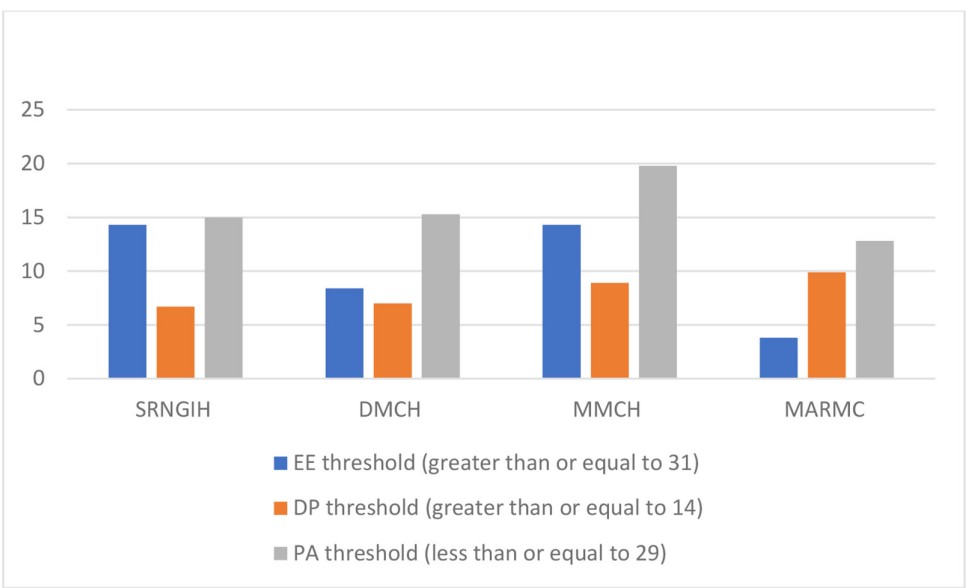

**Figure 2** Proportion of participants in each institution achieving threshold levels for burnout using the MBI scores for EE, DP and PA. Statistically significant results ($\chi^2$ comparison with Bonferroni corrections)—**EE**: SRNGIH vs MARMC, p<0.001; DMCH vs MMCH, p<0.001; DMCH vs MARMC, p<0.002; MMCH vs MARMC, p<0.001. **PA**: SRNGIH vs MARMC, p<0.005; DMCH vs MMCH, p<0.001; DMCH vs MARMC, p<0.002; MMCH vs MARMC, p<0.004. DMCH, Dhaka Medical College Hospital; DP, depersonalisation; EE, emotional exhaustion; MARMC, M Abdur Rahim Medical College; MBI, Maslach Burnout Inventory; MMCH, Mugda Medical College Hospital; PA, personal accomplishment; SRNGIH, The Sheikh Russel Gastro Liver Institute & Hospital.

and burnout,[39] findings that were also mirrored in our study. A recent pilot study in Bangladesh similarly identified a very high prevalence of depression symptoms among Bangladeshi medical students.[26]

Our study has several strengths. We had a large survey sample size of more than 2700 participants with an excellent response rate (90%) that surveyed multiple disciplines and different job roles in four major hospitals in Bangladesh. All information were collected anonymously which may have contributed to the willingness of participants to respond. There were some limitations. The survey tools were administered in English as there was no validated Bengali translation for one of the measures,

and because most of the staff were English speaking. We did however provide some online supplemental information in Bengali to aid completion and added questions at the end of the survey asking for feedback about whether they had understood the survey questions. The responses returned indicated that there were no major problems with comprehension. The high response rate is also likely to indicate that most participants did not experience problems with completion.

The survey itself was limited to doctors and nurses who are likely to have good levels of education. If the survey was rolled out to less qualified staff who may have poorer understanding of the English language, translation into

**Table 3** Hospital Anxiety and Depression Scale (HADS) scores and Maslach Burnout Inventory scores by COVID-19 duty

| | COVID-19 duty | | |
|---|---|---|---|
| **HADS** | **Yes** | **No** | **t-test (p value)** |
| Anxiety Subscale Score | | | |
| Mean (SD) | 8.3 (4.2) | 7.8 (3.9) | 3.4 (p<0.01) |
| Depression Subscale Score | | | |
| Mean (SD) | 7.1 (4.2) | 6.5 (3.7) | 4.2 (p<0.01) |
| **Maslach Subscale Scores** | **Yes** | **No** | **$\chi^2$ (p value)** |
| Emotional exhaustion (EE) | | | |
| Mean (SD) | 17.5 (10.4) | 15.3 (9.8) | 5.4 (p<0.01) |
| Depersonalisation/loss of empathy (DP) | | | |
| Mean (SD) | 4.9 (5.4) | 4.1 (5.0) | 3.9 (p<0.01) |
| Personal accomplishment (PA) assessment | | | |
| Mean (SD) | 37.3 (9.1) | 39.2 (8.1) | −5.4 (p<0.01) |

their native language would be recommended. This survey was conducted during the second wave of COVID-19[40] when stresses on healthcare workers were likely to be high. Further work is needed to explore whether the levels of anxiety, depression and burnout persist in Bangladesh post pandemic.

A further limitation is that our analysis did not control for covariates, in particular, the characteristics of the participants and their awareness of well-being services across institutions. It is possible therefore that the increased levels of anxiety, depression and burnout identified in those more engaged in COVID-19 frontline activities may be due to differences in participant characteristics and/or differences in awareness of well-being services. Future studies should explore the impact (if any) of these factors on levels of anxiety, depression and burnout.

Our findings suggest that working in the healthcare sector in busy and stressful situations may impact on anxiety, depression and burnout. However, as we did not have baseline data from before the COVID-19 pandemic for comparison it is impossible to definitively conclude this. The findings do suggest that healthcare workers may have higher rates than the general population of Bangladesh and prior the COVID-19 pandemic.[41] Healthcare staff levels of anxiety, depression and burnout were worse if they had frontline COVID-19 duties.

Healthcare organisations need to consider how to plan for large-scale events, in future pandemics, to ensure that staff can be appropriately supported during such times while maintaining delivery of high-quality healthcare.

**Author affiliations**
[1]Swansea University, Swansea, UK
[2]Swansea Bay University Health Board, Port Talbot, UK
[3]Cwm Taf Morgannwg University Health Board, Abercynon, UK
[4]Department of Gastroenterology, Sheikh Russel National Gastroliver Institute and Hospital, Dhaka, Bangladesh
[5]Gastroliver Foundation, Dhaka, Bangladesh
[6]Dhaka Medical College and Hospital, Dhaka, Bangladesh
[7]Sheikh Russel National Gastroliver Institute, Dhaka, Bangladesh
[8]Sheik Russel National Gastroliver Institute, Dhaka, Bangladesh
[9]Mugda Medical College and Hospital, Dhaka, Bangladesh
[10]Faculty of Medicine, Health and Life Science, Swansea University, Swansea, UK

**Acknowledgements** The authors thank Jayne Price for her administrative support on the project, staff from Bangladesh involved in data collection/entry and all healthcare professionals who completed the survey.

**Contributors** HAH, JGW, CO'N, KC, GF and MesbahR conceived and secured funding for the study. HAH was the study chief investigator. KC was the research manager and was responsible for overseeing and securing all governance approvals. SI led on data management and statistical analysis with oversight from GF. SR and AJ provided input on study design and interpretation. UD, NH and MesbahR provided clinical input to the study. FA, MH, AKA, MMR, MGK, MasudurR, TM and MA were the Bangladesh principal investigators and were responsible for leading on institutional ethics approvals, site set-up and local recruitment and data collection. HAH led on the drafting of the manuscript. All reviewed and provided input on the final manuscript. HAH is the guarantor for the manuscript and accepts full responsibility for the work and/or the conduct of the study, had access to the data and controlled the decision to publish.

**Funding** This study was funded by Swansea University through their Global Challenges Research Fund. Grant number: N/A.

**Competing interests** None declared.

**Patient and public involvement** Patients and/or the public were not involved in the design, or conduct, or reporting or dissemination plans of this research.

**Patient consent for publication** Not applicable.

**Ethics approval** This study was approved by 1. Swansea University Medical School Research Ethics Sub-Committee (Ref: 2020-0070). 2. Sheik Russel Gastroliver Institute and Hospital Institutional Ethical Review Committee (Ref: SRG&H/ERC/2020-2021/02). 3. M Abdur Rahim Medical College Institutional Review Board (Ref: MARMCD/2021/370). 4. Mugda Medical College & Hospital Institutional Review Board (no ref). 5. Dhaka Medical College Institutional Review Board (Ref: ERC-DMC-ECC/2021/47). Participants gave informed consent to participate in the study before taking part. We provided participants with an information sheet prior to them participating and all were required to sign a written consent form before proceeding with providing questionnaire responses for the survey. Information about participants was kept confidential and managed in accordance with the Data Protection Act, NHS Caldicott Guardian, The Research Governance Framework for Health and Social Care and Research Ethics Committee Approval. Swansea University was the data controller and processor, and we provided training, guidance and practical support to the Bangladesh team, following General Data Protection Regulation (GDPR) guidelines. The project team in Swansea University who collated the data were all trained in GDPR. The only identifiable participant-level information was the signed consent form for the survey which remained in Bangladesh. The paper questionnaires completed in Bangladesh were stored in a locked cabinet owned by one of the Dhaka site principal investigators. The consent forms were held separately to the survey data in a different locked cabinet.

**Provenance and peer review** Not commissioned; externally peer reviewed.

**Data availability statement** Data are available upon reasonable request.

**ORCID iDs**
Hayley Anne Hutchings http://orcid.org/0000-0003-4155-1741
Kymberley Carter http://orcid.org/0000-0003-0691-6282
Ann John http://orcid.org/0000-0002-5657-6995

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
