## [Reviewer comments · BMJ Open]

This paper was submitted to a another journal from BMJ but declined for publication following peer review. The authors addressed the reviewers' comments and submitted the revised paper to BMJ Open. The paper was subsequently accepted for publication at BMJ Open.

ARTICLE DETAILS

TITLE (PROVISIONAL)	Did the COVID-19 pandemic affect levels of burnout, anxiety and depression among doctors and nurses in Bangladesh?: A cross-sectional survey study
AUTHORS	Hutchings, Hayley; Rahman, Mesbah; Carter, Kymberley; Islam, Saiful; O'Neill, Claire; Roberts, Stephen; John, Ann; Fegan, Greg; Dave, Umakant; Hawkes, Neil; Ahmed, Faruque; Hasan, Mahmud; Azad, Abul Kalam; Rahman, Md.; Golam Kibria, Md; Rahman, M. Masudur; Mia, Titu; Akhter, Mahfuza; Williams, John

VERSION 1 – REVIEW

REVIEWER	Baskin, Rachel Villanova University
REVIEW RETURNED	25-Sep-2023

GENERAL COMMENTS	Thank you for submitting your manuscript and for the important work you have done to identify the mental health impacts of the COVID pandemic on healthcare professionals in a LMIC. Please see below for my comments and feedback (also attached in a separate document). The manuscript needs significant work before submitting for publication. There are several concerns I have included that should be addressed. I wish you the best of luck and continued success in your career. • Abstract: anecdotal information is not enough to justify this study. Present data on how COVID impacted the country- what were the death rates/hospitalization rates before or during the time of the study? The introduction fails to adequately describe WHY this study is important.• I am concerned that the authors state in the “already known on the topic” that they are not aware of studies exploring the impact of COVID on LMICs. Since 2020, there have been studies published from India, Singapore, South Africa, etc. documenting this. This shows a lack of knowledge of the extant literature on the topic. It could be said that MOST literature comes from high income countries and there needs to be more literature from LMICs.• Page 6, Row 54: Rather than citing CBS news as a source, go directly to the WHO website and cite their statement.• Page 7, row 25: the management of disorders is not being measured, only the prevalence
--

	 • Page 7, row 36: “Qualitatively the pandemic has caused chaos, but there the impact on people and services has not yet been rigorously measured.” There is no citation to support this statement. Unsure that chaos is the appropriate term. • The methods section does not have any information regarding the participants. How were they recruited? Were specific healthcare professional roles targeted, or any healthcare profession? • The information provided about the measures used in the study questionnaire should be moved from analysis to materials and methods • On page 13, you write “interestingly”, the institution with the highest COVID rota had the worst mental health outcomes. Based on the extant literature, it would be assumed that those staff would have worse mental health outcomes compared to those who did not care for COVID patients. There are no direct research questions or hypotheses in this manuscript, so it is not clear whether these outcomes were anticipated or not. • In the discussion, a study of Bangladeshi nurses conducted during the pandemic is cited, which contradicts earlier statements that there is no data on the impact of COVID in LMIC. • There are several grammar and spelling errors throughout the manuscript. • The figures and tables provide more information on the statistical tests performed than the methods section of the manuscript. • There is no declaration of a lack of conflict of interest.
--	--

REVIEWER	Denning, Max Imperial College London, Department of Surgery and Cancer
REVIEW RETURNED	22-Oct-2023

GENERAL COMMENTS	Overall an excellent paper. It is very impressive to have collected the number of responses and response rate described in this paper. The >90% response rate makes this highly representative of the population and an important contribution to the literature, even more so due to the context of being in an LMIC. A few notes:  1. What was the date range of the survey? Was this contemporaneous or retrospective? A survey conducted in 2020 vs 2021 vs 2022 is likely to have very different impact resulting from Covid - are the feelings due to the excess work following covid-related disruption, or the nature of the work during covid? Depending on the timing, this may lead to some recall bias. This is not necessarily an issue, but will impact the interpretation of the results: if shortly after - that is an interesting snapshot, if these results are later it shows the lasting impact of covid. 2. Was your data normally distributed? Your survey results are reported as means, and analyses used Anova. It is unclear if results were normally distributed - consider if a median or descriptions of categories in the text would be more meaningful than means. 3. In the results section you compare the two groups. In the first 1-2 paragraphs you include details e.g. p values and proportions of staff. Consider including these details in the 3 and 4th paragraph for readability.
--

	4. You have different characteristics across the hospitals e.g. proportion of surgeons, male/female etc. Did you consider a logistic regression or other statistical approach that could control for these differences, and see if the effect of Covid or institution remained? 5. In discussion - consider placing your results in the context of historic findings in Bangladesh. You reference in the introduction section Yasir Arafat et al 2017, and this could provide context. While a direct comparison between your results and 2017 would not be valid, it would give an indication if the Bangladesh health system typically had high rates of burnout, anxiety/depression before the pandemic. 6. In your figures consider showing significance with an asterix/bars as you have conducted these analyses (and reported them in table 2) Other: P7- Try to avoid the term 'poor' in the introduction when describing the population, which is subjective, rather consider using 'low income' or provide reference to a poverty index for an objective measure. p14L28 Bengali spelled as Begali
--	---

VERSION 1 – AUTHOR RESPONSE

Reviewer: 1

Mrs. Rachel Baskin, Villanova University

Comments to the Author:

Thank you for submitting your manuscript and for the important work you have done to identify the mental health impacts of the COVID pandemic on healthcare professionals in a LMIC. Please see below for my comments and feedback (also attached in a separate document). The manuscript needs significant work before submitting for publication. There are several concerns I have included that should be addressed. I wish you the best of luck and continued success in your career.

- Abstract: anecdotal information is not enough to justify this study. Present data on how COVID impacted the country- what were the death rates/hospitalization rates before or during the time of the study? The introduction fails to adequately describe WHY this study is important.

Response: Thank you for these helpful comments. We have now included more information about the impact COVID-19 had in Bangladesh. We have also included why the study is important.

- I am concerned that the authors state in the “already known on the topic” that they are not aware of studies exploring the impact of COVID on LMICs. Since 2020, there have been studies published from India, Singapore, South Africa, etc. documenting this. This shows a lack of knowledge of the extant literature on the topic. It could be said that MOST literature comes from high income countries and there needs to be more literature from LMICs.

Response: We have now amended this text accordingly.

- Page 6, Row 54: Rather than citing CBS news as a source, go directly to the WHO website and cite their statement.

Response: Thank you. We have now included the WHO as the source reference.

- Page 7, row 25: the management of disorders is not being measured, only the prevalence

Response: Thank you. We have now amended this statement.

- Page 7, row 36: “Qualitatively the pandemic has caused chaos, but there the impact on people and services has not yet been rigorously measured.” There is no citation to support this statement. Unsure that chaos is the appropriate term.

Response: Thank you. We have amended this section accordingly and now include references in support of the statements.

- The methods section does not have any information regarding the participants. How were they recruited? Were specific healthcare professional roles targeted, or any healthcare profession?

Response: We have now included more information regarding the participants and how they were recruited in the methods section.

- The information provided about the measures used in the study questionnaire should be moved from analysis to materials and methods

Response: We have now moved this information from the analysis section to the materials and methods section.

- On page 13, you write “interestingly”, the institution with the highest COVID rota had the worst mental health outcomes. Based on the extant literature, it would be assumed that those staff would have worse mental health outcomes compared to those who did not care for COVID patients. There are no direct research questions or hypotheses in this manuscript, so it is not clear whether these outcomes were anticipated or not.

Response: We have now included our research question and a priori hypotheses in the method section.

- In the discussion, a study of Bangladeshi nurses conducted during the pandemic is cited, which contradicts earlier statements that there is no data on the impact of COVID in LMIC.

Response: We have now modified these statements to reflect existing research in LMIC regions.

- There are several grammar and spelling errors throughout the manuscript.

Response: We have now corrected spelling and grammatical errors.

- The figures and tables provide more information on the statistical tests performed than the methods section of the manuscript.

Response: We have now included relevant information regarding statistical tests performed in the methods section as well as the figures/tables.

- There is no declaration of a lack of conflict of interest.

Response: We have now added a conflict of interest statement.

Reviewer: 2

Dr. Max Denning, Imperial College London

Comments to the Author:

Overall an excellent paper. It is very impressive to have collected the number of responses and response rate described in this paper. The >90% response rate makes this highly representative of the population and an important contribution to the literature, even more so due to the context of being in an LMIC.

Response: Many thanks for your positive comments.

A few notes:

1. What was the date range of the survey? Was this contemporaneous or retrospective?

A survey conducted in 2020 vs 2021 vs 2022 is likely to have very different impact resulting from Covid - are the feelings due to the excess work following covid-related disruption, or the nature of the work during covid? Depending on the timing, this may lead to some recall bias. This is not necessarily an issue, but will impact the interpretation of the results: if shortly after - that is an interesting snapshot, if these results are later it shows the lasting impact of covid.

Response: We have added the date range of the survey to the manuscript.

2. Was your data normally distributed? Your survey results are reported as means, and analyses used Anova. It is unclear if results were normally distributed - consider if a median or descriptions of categories in the text would be more meaningful than means.

Response: We took the decision to apply parametric tests and statistics as the sample size was sufficiently large for the Central Limit Theorem to apply and as such that parametric tests (including the ANOVA) were appropriate.

3. In the results section you compare the two groups. In the first 1-2 paragraphs you include details e.g. p values and proportions of staff. Consider including these details in the 3 and 4th paragraph for readability.

Response: Thank you. We have now included these details in the 3rd and 4th paragraph.

4. You have different characteristics across the hospitals e.g. proportion of surgeons, male/female etc. Did you consider a logistic regression or other statistical approach that could control for these differences, and see if the effect of Covid or institution remained?

Response: Thank you for this suggestion. We had a large amount of data on individual characteristics. As our a priori research question was whether healthcare staff working on covid rotas had worse anxiety, depression and burnout than those staff not on covid rotas, we chose to focus on these comparisons at this stage rather than applying analyses such as logistic or linear regression which would focus more on factors that may contribute to the burnout, stress and anxiety scores. We also have a second aligned study that assessed workload burden within each institution at the time. We plan to use the information from this study to undertake more detailed analysis of the data.

5. In discussion - consider placing your results in the context of historic findings in Bangladesh. You reference in the introduction section Yasir Arafat et al 2017, and this could provide context. While a direct comparison between your results and 2017 would not be valid, it would give an indication if the Bangladesh health system typically had high rates of burnout, anxiety/depression before the pandemic.

Response: Thank you. We have now referenced our findings in relation to the appropriate papers. The Yasir Arafat paper from 2017 did not measure burnout so we have therefore reworded the statement to clarify this.

6. In your figures consider showing significance with an asterisk/bars as you have conducted these analyses (and reported them in table 2)

Response: We have now indicated with an asterisk, where significant results exist.

Other:

P7- Try to avoid the term 'poor' in the introduction when describing the population, which is subjective, rather consider using 'low income' or provide reference to a poverty index for an objective measure.

Response: Thank you for this helpful suggestion. We have now removed the term poor and changed this to low income. We have also added a reference to the poverty index.

p14L28 Bengali spelled as Begali

Response: This is now corrected.

Reviewer: 1
 Competing interests of Reviewer: None

Reviewer: 2
 Competing interests of Reviewer: Nil

VERSION 2 – REVIEW

REVIEWER	Baskin, Rachel Villanova University
REVIEW RETURNED	20-Dec-2023

GENERAL COMMENTS	 • Abstract, methods section: remove the parentheses from before Hospital Anxiety and just incorporate the name of the scales into the sentence • Since COVID is an abbreviation, should it be capitalized? • In opening sentence from introduction, “Bangladesh is a Low and/or Middle Income Countries (LMIC) country”- rather than stating country twice, rephrase the sentence to read better. • The information on the measures included in the study was well written. Perhaps adding in reliability (i.e., Cronbach alpha) from either the current study or previous studies would be helpful in demonstrating the reliability. Thank you for including that you used English versions of the tools and the rationale. • Discussion section is well written and provides a good overview of the findings and implications for future work. • There are some small grammar errors throughout the manuscript that need to be addressed, such as missing commas.
--

REVIEWER	Denning, Max Imperial College London, Department of Surgery and Cancer
REVIEW RETURNED	03-Jan-2024

GENERAL COMMENTS	P17 Para 1 - I disagree with the conclusion. The authors can only comment on the comparisons drawn. Without describing the baseline of general rates of depression in Bangladesh, it cannot be concluded that “working in the healthcare sector in busy and stressful situations has a negative impact on anxiety, depression and can increase levels of burnout in healthcare workers in Bangladesh”. It can be described that for example, “there is a high rate of XYZ, which is higher than comparable studies of the general population”. I do agree with the interpretation that staff with frontline Covid responsibilities had worse rates of Anxiety/depression/burnout. Discussion The aim of the study is to explore levels of burnout, anxiety and depression across healthcare workers. While the study does answer this questions, in the discussion, there are topics for additional analysis or study that could be suggested: Given there are clear differences in the characteristics across institutions, it should be noted as a limitation that the analysis has not controlled for covariates. Therefore it remains speculative that the differences in Covid Rota may have lead to the differences in anxiety/depression/burnout across institutions. For example MMCH has the worst scores for anxiety/depression/burnout but also the highest proportion of doctors and the lowest awareness of
--

	the wellbeing service. Both of these could also be plausible explanations for differences in psychological wellbeing. There has also been no comment on differences across institutions in the awareness of/access to wellbeing services. Given the conclusion is “These findings could help healthcare organisations to help plan for future similar events.” The authors should consider commenting on whether the presence/awareness of these services (a modifiable factor) has any association with psychological wellbeing. This would also represent a topic of interest for future research. Study description: The authors repeatedly describe the study as “cross sectional prospective”. As it is a single time point (cross sectional), it should not also be described as prospective (which implies longitudinal) unless additional measures were conducted Figures: In the response to reviewers, the authors state they have updated the graphs to show significance. This is not present on the version I have received. Figure 2 - unclear what this is showing, is this the proportion of participants above the threshold? Add ‘%’ to Y axis, or clarify with a caption or change the legend so that the word ‘threshold’ is within the parentheses Table 1 - in the ‘Profession’ row there is a second line after ‘other’ - what does this correspond to? Is this missing data? Clarify ‘Wellbeing service’ is this - aware of or having accessed? The distribution of respondents seems surprising for SRNGIH. >50% of the respondents are based in the gastroenterology department. In the methods, the authors state they surveyed the entire hospitals and received response rates of 90%. Suggest that the authors confirm these data. Additional comments: 6 L42 - It was at risk (vs potential to be a risk candidate) P6 in general - consider tenses P7 L47-60 - Don’t just describe that work was undertaken, allude to the findings. P8 Aim - add ‘during [±following] the Covid-19 pandemic’ P16 L 45-47 You describe the results as limited to doctors and nurses. but in the methods you describe “healthcare staff working across all specialties” and Table 1 you have ‘Other’ under ‘professions’
--	--

VERSION 2 – AUTHOR RESPONSE

Reviewer: 1

Mrs. Rachel Baskin, Villanova University

Comments to the Author:

- Abstract, methods section: remove the parentheses from before Hospital Anxiety and just incorporate the name of the scales into the sentence

Response: We have removed the parentheses from this sentence.

- Since COVID is an abbreviation, should it be capitalized?

Response: We are aware that there is currently some debate about whether this should be capitalized. We have adopted the WHO position with regards to this and have used capitals throughout (20200211-sitrep-22-ncov.pdf (who.int)). We are however content to be guided by the journal editorial team and can change this to Covid-19 if this is the preference.

- In opening sentence from introduction, “Bangladesh is a Low and/or Middle Income Countries (LMIC) country”- rather than stating country twice, rephrase the sentence to read better.

Response: We have rephrased this sentence.

- The information on the measures included in the study was well written. Perhaps adding in reliability (i.e., Cronbach alpha) from either the current study or previous studies would be helpful in demonstrating the reliability. Thank you for including that you used English versions of the tools and the rationale.

Response: We have now added in details about reliability of both the scales used including references.

- Discussion section is well written and provides a good overview of the findings and implications for future work.

Response: Thank you. We appreciate the time you have taken to review the manuscript.

- There are some small grammar errors throughout the manuscript that need to be addressed, such as missing commas.

Response: We have now re-read the manuscript and corrected grammatical errors.

Reviewer: 2

Dr. Max Denning, Imperial College London

Comments to the Author:

P17 Para 1 - I disagree with the conclusion. The authors can only comment on the comparisons drawn. Without describing the baseline of general rates of depression in Bangladesh, it cannot be concluded that “working in the healthcare sector in busy and stressful situations has a negative impact on anxiety, depression and can increase levels of burnout in healthcare workers in Bangladesh”. It can be described that for example, “there is a high rate of XYZ, which is higher than comparable studies of the general population”. I do agree with the interpretation that staff with frontline Covid responsibilities had worse rates of Anxiety/depression/burnout.

Response: We have now modified our conclusions based on this feedback.

Discussion

The aim of the study is to explore levels of burnout, anxiety and depression across healthcare workers. While the study does answer these questions, in the discussion, there are topics for additional analysis or study that could be suggested:

Given there are clear differences in the characteristics across institutions, it should be noted as a limitation that the analysis has not controlled for covariates. Therefore it remains speculative that the differences in Covid Rota may have led to the differences in anxiety/depression/burnout across institutions. For example MMCH has the worst scores for anxiety/depression/burnout but also the highest proportion of doctors and the lowest awareness of the wellbeing service. Both of these could also be plausible explanations for differences in psychological wellbeing.

There has also been no comment on differences across institutions in the awareness of/access to wellbeing services. Given the conclusion is “These findings could help healthcare organisations to

help plan for future similar events.” The authors should consider commenting on whether the presence/awareness of these services (a modifiable factor) has any association with psychological wellbeing. This would also represent a topic of interest for future research.

Response: Thank you for these comments. We have now amended the discussion accordingly and added further discussion around limitations and future work. We have also added these points to the strengths and limitations section.

Study description:

The authors repeatedly describe the study as “cross sectional prospective”. As it is a single time point (cross sectional), it should not also be described as prospective (which implies longitudinal) unless additional measures were conducted.

Response: We have now removed the term prospective when describing the survey.

Figures:

In the response to reviewers, the authors state they have updated the graphs to show significance. This is not present on the version I have received.

Responses: Apologies for this omission. Many of the comparisons are significant so rather than identify them all in the graph (which resulted in a more cluttered and confusing version), we have added details of significant results as a footnote.

Figure 2 - unclear what this is showing, is this the proportion of participants above the threshold? Add ‘%’ to Y axis, or clarify with a caption or change the legend so that the word ‘threshold’ is within the parentheses

Response: Thank you for this feedback. We have amended the figure accordingly.

Table 1 - in the ‘Profession’ row there is a second line after ‘other’ - what does this correspond to? Is this missing data?

Response: Apologies- yes this is missing data. We have now corrected the table accordingly.

Clarify ‘Wellbeing service’ is this - aware of or having accessed?

Response: This is being aware of wellbeing services. We have added ‘awareness’ to Table 1.

The distribution of respondents seems surprising for SRNGIH. >50% of the respondents are based in the gastroenterology department. In the methods, the authors state they surveyed the entire hospitals and received response rates of 90%. Suggest that the authors confirm these data.

Additional comments:

Response: We have rechecked these data and they are correct. SRNGIH focuses largely on gastroenterology/liver services and as such the number of respondents is more heavily weighted towards gastroenterology.

6 L42 - It was at risk (vs potential to be a risk candidate)

Response: We have corrected ‘potential to be a risk candidate’ to ‘at risk’.

P6 in general - consider tenses

Response: We have now thoroughly proofread the manuscript and changed tenses where required.

P7 L47-60 - Don’t just describe that work was undertaken, allude to the findings.

Response: We have now added details of the findings from these studies.

P8 Aim - add ‘during [±following] the Covid-19 pandemic’

Response: We have added during the COVID-19 pandemic to the aim.

P16 L 45-47 You describe the results as limited to doctors and nurses. but in the methods you describe "healthcare staff working across all specialties" and Table 1 you have 'Other' under 'professions'

Response: This was directed at doctors and nurses and as such have changed 'healthcare staff' to doctors and nurses. 'Other' under professions category is for trainee nurses and doctors. We have now amended this accordingly.

Reviewer: 1

Competing interests of Reviewer: None

Reviewer: 2

Competing interests of Reviewer: Nil